# Interdisciplinary Children’s Behavioral Health Workforce Development for Social Work and Nursing

**DOI:** 10.3390/ijerph20085601

**Published:** 2023-04-20

**Authors:** Elizabeth Palley, Chireau White, Chrisann Newransky, Marissa Abram

**Affiliations:** 1School of Social Work, Adelphi University, Garden City, NY 11530, USA; 2College of Nursing and Public Health, Adelphi University, Garden City, NY 11530, USA

**Keywords:** children’s mental health, workforce development, behavioral health, social work training, nurse practitioner training, interdisciplinary education

## Abstract

This paper will begin with a review of child health inequities globally, in the United States and in the State of New York. It will then describe a model training program that was designed to educate social workers and nurse practitioners to create a workforce able to address child behavioral health inequities in the United States (US), specifically New York State. Behavioral health care refers to prevention, care and treatment for mental health and substance abuse conditions as well as physical conditions caused by stress and life crises. This project uses an interdisciplinary training program for nurse practitioner and Master of Social Work students to address workforce shortages in underserved communities in New York State. It will present process evaluation findings to highlight the program’s initial success and will conclude with a discussion of the data that are still needed and the challenges of obtaining this data.

## 1. Introduction

Worldwide, it is estimated that 1 in 7 (14%) of 10–19 year-olds experience mental health conditions, yet these remain predominantly under-recognized and undertreated [1]. According to the United Nations Children’s Fund (UNICEF) [2], “these barriers are systemic blocks established by a lack of funding, leadership, coordination among sectors and trained workers”. For example, in low- and middle-income countries, mental health service rates fall below 1 per 100,000 people, whereas in high-income countries the rate is 1 per 2000 people [3]. The United Nations Sustainable Development Goals were created in September 2015, with the intent to “mobilize efforts to end all forms of poverty, fight inequalities and tackle climate change, while ensuring that no one is left behind” [4]. Goal 10 highlights the importance of reducing inequality within and among countries. According to the World Health Organization [3], “depression, anxiety and behavioral disorders are among the leading causes of illness and disability among adolescents”. In 2021, the UNICEF State of the World’s Children put out a call to action to promote the mental health of children and adolescents, reporting that the investment in youth mental health was negligible. The report highlighted how the cost of inaction with mental health promotion is a significant barrier to human potential and contributes to the disruption of youth meeting developmental milestones. To address this issue, UNICEF recommended investment in mental health services and workforce development along with evaluation that centered around robust data collection and research [2].

In the US in any given year, there are 1 in 7 children between the ages of 3–17 years experiencing a behavioral health disorder. Yet, according to a recent analysis of population data, only about half receive behavioral health services to treat their conditions [5]. Most recently, Lisa Conrad of The Commonwealth Fund reported that there was a 43% increase in pediatric mental health visits to emergency rooms between 2015 and 2020, 13% of whom return within 6 months of their initial visit [6]. In 2018, the New York State Office of Mental Health (OMH) highlighted the importance of access to behavioral health for children, adolescents and transitional age youth [7]. According to the report, increasing access to behavioral health especially in primary care settings may lead to early identification and intervention, which in turn can result in better outcomes for individuals, families and communities. These outcomes, according to the OMH 2018 report, may include reducing the risk of children ending up in the juvenile justice or other child-serving systems. Additionally, behavioral health disorders, such as depression, are a risk factor for suicide. While the suicide death rate is lower among youth than other age cohorts, a 2016 OMH report on suicide reported that the number of suicide attempts among those aged 15–24 is much higher now than in the past and is now the second leading cause of death in that age group [8].

While there are many barriers to the receipt of behavioral health services, in 2020, the Citizens’ Committee for Children of New York (CCCNY) [9] reported that adequate access to care is further exacerbated by the lack of trained behavioral health professionals. The Health Resources Services Administration (HRSA), a federal agency tasked with increasing health equity by providing care to geographically, economically isolated and medically vulnerable populations, projects that the national supply of behavioral health professionals will meet the 12–15% increase in demand between 2016 and 2030; however, these projections do not account for uneven regional distributions of the workforce and the different care delivery models [10]. In part, the increase in demand for behavioral health services will be met with an increase in the use of social workers. The Mullin Institute at George Washington University in collaboration with the National Association of Social Workers (NASW) [11] found that sixty percent of recent graduates (up to 3 years post Master of Social Work) were employed providing behavioral health care. It is not clear what type of education and training in integrated behavioral healthcare they received in their respective educational programs or if they received specialized training in treating the behavioral health needs of children, adolescents and transitional-age youth. Interestingly, the HRSA projected shortage of behavioral health care providers is based on the age of existing providers and the growing need for behavioral health care [12].

Psychiatric workforce shortages remain a serious issue in the field of behavioral health. Currently, there are 5112 mental health professional shortage areas (HPSAs) across the US designated by HRSA [13]. These are areas where mental health care access is more limited than in the rest of the country. HRSA estimates 45,580 psychiatrists, 7670 psychiatric mental health nurse practitioners, and 1280 behavioral health physician assistants (PAs) are currently practicing in the U.S. The geographic distribution of these professionals is uneven. Most are located in the Northeast and Pacific Northwest, thereby leaving a significant proportion of the country without adequate coverage. The uneven distribution of behavioral health professionals is also seen in New York State, specifically in pockets in the Bronx, Brooklyn, Westchester, the Hudson Valley and parts of Long Island.

The outbreak of the COVID-19 pandemic put a spotlight on and exacerbated the behavioral health needs of children worldwide and in New York State. According to the Citizens’ Committee on Children of New York (CCCNY), 54.5% of children with behavioral health needs in New York did not receive needed treatment in 2019–2020 (CCCNY, 2020). Despite the disruptions to in-person services resulting from the pandemic, temporary changes in health care regulations enabled many behavioral health providers to respond to behavioral health needs by providing telehealth behavioral services. Still, some challenges of access remain, especially at the client-level. Mace et al. [14] identified the lack of technology and health insurance among the barriers faced by clients in fully utilizing telehealth services. Other researchers have identified limited literacy and stigma associated with mental health as additional barriers at the agency and provider level [15,16,17].

In the last two decades, diversity, equity, and inclusion (DEI) has been an area of development in healthcare [18]. Black, Indigenous and people of color (BIPOC) have historically been underrepresented in the healthcare workforce [19] (Children of underserved and historically marginalized groups often experience poor mental health outcomes that result from socioeconomic status, adverse childhood experiences and access to community resources [20,21,22]. In a recent report from the Center for Disease Control and Prevention, “suicide rates among persons aged 10–24 years increased significantly during 2018–2021 among individuals who identified as Black” [23]. Among high school aged youth, female students, LGBTQ+ students and students who had any same-sex partners were more likely than their peers to experience poor mental health, suicidal thoughts and suicidal behaviors [24]. Children from marginalized groups also contend with lack of access to culturally appropriate services [25,26,27,28]. With over 30% of residents in New York speaking a language other than English, there is a need for a behavioral health workforce that can address the cultural and linguistic needs of children [17,25,28,29,30,31,32]. In the fields of social work and nursing, representation of BIPOC social workers and nurses have continued to expand in comparison to other helping professions in hopes of meeting these needs and reducing inequalities among child, adolescent and transitional-age youth [33]. However, progress is still needed.

## 2. Materials and Methods

The Interdisciplinary Education and Training (IDEATE) program was created as a result of a HRSA-funded (Health Resources and Services Administration) federal training grant for professional schools to train behavioral health practitioners to work with children working in interdisciplinary settings in the United States. Our project had two main goals: create a comprehensively, interdisciplinarily trained, diverse behavioral health workforce of social workers and nurse practitioners with a specific emphasis on the provision of integrated behavioral health in primary care settings to children, adolescents and transitional-age youth; and second, to increase access to high quality integrated behavioral healthcare in underserved and high demand communities. To guide us in this endeavor, we created a logic model that would identify underlying assumptions and outline the inputs, activities, outputs, and outcomes that would lead to the impact we are expecting from this program (see Appendix A). This was not a randomized control study, and, therefore, the findings are limited because there may have been selection bias in the students who chose to apply and participate in the program. In addition, we did not survey students who did not participate in the program on the study measures. In the first year of this grant, 2021–2022, we trained 28 graduate students, 15 psychiatric nurse practitioner students and 13 Master of Social Work (MSW) students. In the second year, 2022–2023, we have been training 32 students, including 19 MSW students and 13 psychiatric mental health nurse practitioner (PMHNP) students.

The following highlights some areas of the logic model.

### 2.1. Assumptions

In this training model, we had two major assumptions based on the reviewed literature and data for our state. The first was that a trained, diverse and interdisciplinary workforce in integrated behavioral health can better address the needs of children, adolescents and youth in HPSAs in New York State. The second assumption was that a cohort model allows students to mutually support and learn from each other via shared experiential and didactic learning.

### 2.2. Inputs

Faculty and administrators at Adelphi University’s School of Social Work and the College of Nursing and Public Health worked together to create experiential training opportunities for approximately 30 advanced level MSW and PMHNP students per year (or 120 students in 4 years) who have and will specifically focus on integrated behavioral health treatment with children, adolescents and transitional-age youth. While social workers in the United States generally provide therapy and case management and are trained in macro level advocacy, psychiatric mental health nurse practitioners are trained to assess and diagnose mental health care conditions and can prescribe medication for both mental health and substance abuse disorders [34,35]. In this model, student trainees have two full semesters (30 weeks) of didactic curriculum, specialized seminars and experiential training in integrated behavioral health in primary care or behavioral health settings by faculty, administrators and agency partner experts.

### 2.3. Activities and Outputs

In order to create this training, we reviewed and revised existing course pathways through both graduate programs. Though most of the coursework is identical to other nurse practitioner and social work master level students, social work students in the IDEATE program are required to take specific electives including Social Work for Children and Adolescents, Social Work in Practice in Health Care and Evidence Based Practice with Severe Mental Illness. The nurse practitioner program was not altered as it was deemed to include the necessary coursework to prepare students to work in integrated behavioral health settings. We worked with existing agency partners and created new partners to place students in primary care and behavioral health settings for the practicum component of the social work curriculum and the clinical component of the nursing curriculum. Second year MSW students are required to complete 450 hours in a social work site based practicum. Nurse practitioner students must complete 200 clinical placement hours over the course of the program.

In addition to the core curriculum, we created educational workshops that both the nursing and social work students participated in together. This online professional development series has used case studies, conferences, simulations and cutting edge topics including evidence-based approaches to suicide prevention, collaborative care, tele-play therapy techniques and digital health innovations in behavioral health. In these workshops, we have also worked on understanding and addressing health disparities, clinical documentation, digital mental health literacy, substance use disorder, leadership in healthcare as well as best practices and skills in the provision of remote behavioral health services.

To address the challenges such as limited interdisciplinary clinical placements, our training has included virtual clinical simulations. Our first clinical simulation used two case scenarios, centered around depression, anxiety and suicide. These topics were chosen based on a review of the literature, previous seminar content and commonly seen symptoms across clinical settings of our students. Four student actors were recruited from Adelphi University’s theater department and trained by faculty to represent the patients in these cases. Due to COVID-19 and preferences expressed by the students, the simulation was held via Zoom (Zoom Video Communications, Inc., San Jose, California, USA). The clinical simulation structure was comprised of (1) large group introductions, overview of simulation and instructions on format; (2) students breaking up into small groups, based on PMHNP and MSW pairings; (3) pairs participating in simulation with faculty present for observation using a check-list; (4) students participating in a focus group to provide feedback on their experience; and (5) large group wrap-up and administration of mid survey scale.

Based on focus group feedback and faculty observation of simulations, we decided that another clinical simulation would be advantageous for students. The second simulation focused on substance abuse and was held in the spring semester. We identified technology, preparation and need for debriefing as areas to improve for the second simulation. We also identified several opportunities for additional education, such as (1) collaboration between nurse practitioner and social work teams; (2) legal rights of minors and parents in clinical care planning; (3) prioritizing risk assessment during evaluation; and (4) integrating screening tools into evaluation/assessment.

We also had someone from the University career services office attend one of the final workshops to help give students support with job searches and applications.

We have evaluated the program so far largely based on student reports. We used a modified version of Davis’s Social Worker Integrated Care Competencies Scale (SWICCS) and Phillips Objective Structured Clinical examination (OSCE) Telehealth Survey [36,37]. The survey contains 48 items, using a rating scale of 1 (low) to 5 (high). It is composed of 9 sections to assess student knowledge in Integrated Care, Assessment and Diagnosis, Care Coordination, Diversity, Documentation, Healthcare Basics, Evidence Based Practices, Technology and Telehealth. Survey questions focused on a student’s level of confidence in knowledge and understanding of each domain. For the Integrated Care domain, students were asked about their knowledge of behavioral health care, their understanding of the connection between integrated care and health inequities, knowledge of professional roles and skills specific to integrated care and confidence in collaborating within an interdisciplinary team. For the Assessment and Diagnosis domain, students were asked about their ability to identify, administer and interpret screening tools, use appropriate assessment based probing questions, formulate a diagnosis based on the Diagnostic and Statistical Manual of Mental Disorders Fifth Edition (DSM-5) criteria and prescribe medication based on clinical impressions. For the Care Coordination domain, students were asked about their understanding of linkages to holistic resources (social, medical and mental health), ability to interpret records notes from different disciplines for treatment planning and preparation of case summaries for interdisciplinary teams. For the Diversity domain, students were asked to assess their knowledge of culturally diverse populations, ability to incorporate culture into treatment planning, build rapport with diverse populations, utilize language interpreter services and apply anti-racist principles to patient care. For the Documentation domain, students were asked about their ability to document professionally, clearly, concisely and in a timely manner. For the Healthcare Basics section, students were asked about their knowledge of common health conditions and common medications, appropriate preparation of notes for integrated care setting and effective communication around co-occurring disorders. For the Evidence Based Practices domain, students were asked about their knowledge of several evidence-based interventions that include motivational interviewing, cognitive behavioral therapy, dialectical behavioral therapy, solution focused therapy, trauma informed care and crisis intervention. For the Technology domain, students were asked about their ability to navigate an electronic health record and use technology, such as computers and mobile devices, for patient care. Lastly, the Telehealth domain asked students to reflect on their confidence to build rapport with a patient, collect patient history and communicate effectively using a telehealth platform. Students in our first cohort were surveyed at the outset, mid-year and at the end of the program. We also had students participate in a focus group in June 2022. Students are projected to complete a final post survey one year after completing the program. Preliminary program outcomes and discussion of the impact follow in the Results and Discussion sections below.

## 3. Results

Our preliminary data suggests that students felt the program was helpful. In terms of competencies, though students began with the lowest levels of confidence across all survey domains, scores improved significantly both at mid-year and at the end of year evaluation. We only have midyear evaluation data for two years of students, so we have relied on that here instead of end of year data which we only have for one year.

We used SPSS to analyze data obtained from the survey. We used a paired sample t-test to compare mean scores between pre- and mid-year and pre- and post-year. As seen in Table 1, pre- and mid-year mean findings showed that students perceived that their knowledge improved across all domains but with greater change in the following: assessment and diagnosis (pre- 2.41, mid-year 3.41, *p* < 0.01), care coordination (pre- 2.69, mid-year 3.52, *p* < 0.01), health care basics (pre- 2.52, mid-year 3.46, *p* < 0.01), and evidence based practice (pre- 2.25, mid-year 3.17, *p* < 0.01). As seen in Table 2, pre- and post-year findings showed that students perceived that their knowledge improved across all domains. Integrated care (pre- 3.06, post-year 4.57, *p* < 0.001), assessment and diagnosis (pre- 2.41, post-year 3.91, *p* < 0.001), care coordination (pre- 2.69, post-year 4.25, *p* < 0.001), diversity (pre- 3.44, post-year 4.8, *p* < 0.001), documentation (pre- 3.29, post-year 4.44, *p* < 0.001), health care basics (pre- 2.52, post-year 4.17, *p* < 0.001), evidence based practice (pre- 2.25, post-year 4.08, *p* < 0.001), technology (pre- 3.20, post-year 4.44, *p* < 0.001), and telehealth (pre- 3.99, post-year 4.81, *p* < 0.001). In other words, students reported statistically significant improvements in all areas of learning that we assessed.

## 4. Discussion

This study examined a new method of training social worker and nurse practitioner students which involves interdisciplinary collaboration. We have plans to evaluate and follow up with students approximately one year after they have completed the program, but that is, obviously, not indicative of a long term commitment to children’s behavioral health. It will just indicate how they are starting their post-graduate careers. We also need to know how many of our students end up working in underserved areas with children/adolescents and, if so, for how long? Ideally, we would be able to follow up with our graduates continually on an annual basis. In addition, to truly learn how effective the program is, it would be important to be able to assess the health status of the target population to ensure that it has improved as a result of our training. It might also be helpful to learn about the innovations our graduates have brought to their agencies. Are our graduates identifying problems and creating programs and innovations to meet the needs of the target population? Are they working as advocates to change policy at the local, state or federal level to enable better coordination and access to behavioral health care? Though we are obligated to report student outcomes one year after leaving the program, our graduates may be hard to reach on an ongoing basis. Ultimately, we would like to know if our program leads to a reduction in behavioral health inequities. 

There are several different study designs we could use to better assess our ability to meet the program goals: (1) quasi experimental comparison study, (2) a five-year mixed methods longitudinal study, (3) community based research, or (4) impact evaluation. A quasi experimental study would involve recruiting nurse practitioner and social work students who were not in the fellowship and have graduated as a control group. We would administer a post survey alongside the one-year post to our cohort 2 IDEATE students. If we were able to obtain this data, we could better understand whether training through our fellowship provides an edge in the development of student competencies for healthcare settings that focus on underserved children and families. We would also be able continue this comparison model for future cohorts. A second option could be the use of a five-year mixed method longitudinal study. It would allow for us to monitor progression of competencies using the Social Worker Integrated Care Competencies Scale and Phillips OSCE Telehealth Survey. It would also provide an opportunity to assess retention and application of fellowship content over time. Additionally, it would allow for individual interviews or small focus groups from cohort 1 and 2 to better capture information that quantitative instruments are unable to measure. A third option would be the use of community-based research targeting the supervisors and patients with whom our students work to understand the facilitators and barriers to students achieving relevant competencies. Additionally, we would be able to obtain input on whether the instruments we are using with students are appropriately covering the types of evidence based interventions needed for clients. We would also learn more on whether fellowship content aligns with the clinical skills being taught at their practicum site. The last option we could explore is an impact evaluation study. An impact evaluation study would focus on the effectiveness in achieving our program’s ultimate goals. It would require contacting stakeholders (students, supervisors, directors, patients, field liaisons, IDEATE team members), interviewing stakeholders to help determine whether we have achieved program success, assessing effectiveness of our measures, evaluating our ability to place students in internship settings that function in an interdisciplinary model and an internal review on whether existing processes have been working towards program goals. Although many of these research methods would broaden our ability to track and monitor program and student outcomes, some would also be time-consuming and expensive.

Lastly, for this program to really make a difference, continued work with programs, such as IDEATE, must be sustainable without federal grant funding, resources and infrastructure, such as the infrastructure that already existed around nurse practitioner and social worker training at Adelphi University before this grant was started. They must also be transferable outside of the United States.

## 5. Conclusions

The IDEATE training program was designed to improve interdisciplinary collaboration between nurses and social workers and to train MSW and nurse practitioner students to better work together and provide behavioral health care to children, adolescents and transitional-age youth. It is in alignment with UNICEF’s call to action to improve and increase behavioral health services for children and adolescents and integrates novel approaches to provide interdisciplinary training through high-impact learning strategies built into the professional development seminars and clinically focused simulations. Though our students have so far reported that the program has provided them with useful skills that have enabled them to do this work during their practicum, we need to collect more outcome data. Furthermore, for the training to have a significant impact, trainees would have to remain in the field of children’s behavioral health care in underserved communities; if not, then the program will not, ultimately, be successful.

## Figures and Tables

**Table 1 ijerph-20-05601-t001:** Pre and Mid Mean Scores by Student Competency Domains.

Domains	Pre-	Mid-	*p* Values
Integrated Care	3.06	4.04	<0.001
Assessment and Diagnosis	2.41	3.41	<0.001
Care Coordination	2.69	3.52	<0.001
Diversity	3.44	3.89	0.007
Documentation	3.29	3.75	0.002
Healthcare Basics	2.52	3.46	<0.001
Evidence Based Practices	2.25	3.17	<0.001
Technology	3.2	4.01	<0.001
Telehealth	3.99	4.38	0.004

Notes: *n* = 28 students. Student competency domains adapted from the Social Worker Integrated Care Competencies Scale (SWICCS) and Phillips Objective Structured Clinical Examination (OSCE) Telehealth Survey.

**Table 2 ijerph-20-05601-t002:** Pre and Post Mean Scores by Student Competency Domains.

Domain	Pre-	Post-	*p* Values
Integrated Care	3.06	4.57	<0.001
Assessment and Diagnosis	2.41	3.91	<0.001
Care Coordination	2.69	4.25	<0.001
Diversity	3.44	4.38	<0.001
Documentation	3.29	4.44	<0.001
Healthcare Basics	2.52	4.17	<0.001
Evidence Based Practices	2.25	4.08	<0.001
Technology	3.2	4.44	<0.001
Telehealth	3.99	4.81	<0.001

Notes: *n* = 28 students. Student competency domains adapted from the Social Worker Integrated Care Competencies Scale (SWICCS) and Phillips Objective Structured Clinical Examination (OSCE) Telehealth Survey.

## Data Availability

Data available on request. Please contact Chrisann Newransky at cnewransky@adelphi.edu.

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
