# Peer review of "Interdisciplinary Children’s Behavioral Health Workforce Development for Social Work and Nursing"

_ijerph, 2023, doi:10.3390/ijerph20085601_

Round 1
Reviewer 1 Report
Reviewer comments and suggestions
The authors in this study discussed a model training program designed to educate social workers and nurse practitioners to create a workforce able to address child behavioral health inequities in the United States (US), specifically in New York State.
Behavioral health care refers to prevention, care and treatment for mental health and substance abuse conditions as well as physical conditions caused by stress and life crises They used an interdisciplinary training program for nurse practitioner students and MSW students to better prepare these students for their workforce challenges.
Overall, the manuscript was well written. However, a few concerns/comments needed to be explained/modified.
- Line 7-8 Can we write New York city of United states.
- Line 13 MSW should be in full form and a typo error in line 14 was seen
- Line 15-16 The last line should be modified, there should be some message present by your manuscript.
- Line 98 and several places no need to add the reference et al if the authors already put the reference number, please delete it in the whole MS
- Line 124-129 How many participants the authors covered for this study, It should be mentioned here not in other section
- Line 153-154 In my thinking a ray diagram or that kind of explanation is important to easily understand your manuscript. Try to prepare it for reader-friendly of your MS
- Table 1 These domains should be explained well
- First para of discussion Better not to include these statements in the first para, try to mention the novelty of this study
- Line 265 is this necessary to pose a question, better to solve the query is a best idea for the common reader of your paper.
- All reference formats should be modified based on MDPI guidelines.
Author Response
In response to the feedback
1) We added "State of" to New York
2) We wrote out Masters of Social Work and deleted the end of the sentence.
3) We changed the last sentence to "It will provide some initial process outcome data to highlight the program's success and will conclude with a discussion of data that is still needed and the challenges of obtaining this data." The message is that we are sharing a successful program that was designed to create a workforce designed to address children's mental health.
4) In-text references were deleted from the manuscript.
5) The number of participants was moved to this section from the previous section.
6) Because the only curricular changes were the addition of two electives to the social work program, we thought it would be confusing to include a ray diagram with the names of classes in both the Masters of Social Work and Nurse Practitioner programs. Also, the class names would not have much meaning without including the curriculum. This did not seem appropriate for this paper.
7) The domains in the table were explained on page 5, line 217/218 when they were initially discussed.
8) The first paragraph of the discussion was changed to "This study examined a new method of interdisciplinary training for social worker and nurse practitioner students which involves interdisciplinary collaboration." I deleted the second sentence.
9) Line 265 was rewritten as a declarative sentence.
10) The references were modified based on MDPI guidelines.
Reviewer 2 Report
1. Suggest adding reference to review of Commonwealth Foundation of state activities to promote outreach services in adolescent mental health, including funding under Medicaid. Available at: MARCH 8, 2023
Strengthening Home- and Community-Based Services to Stabilize Young People with Behavioral Health Problems and Keep Them Out of Hospitals
Laura Conrad
Senior Consultant, Technical Assistance Collaborative, Inc.
CITATION
Laura Conrad, “Strengthening Home- and Community-Based Services to Stabilize Young People with Behavioral Health Problems and Keep Them Out of Hospitals,” To the Point (blog), Commonwealth Fund, Mar. 8, 2023. https://doi.org/10.26099/60tc-m972
2. Discussion would benefit from added para of recommendations for follow-up studies of benefit of use of culturally appropriate Community Health Workers (CHWs) for outreach as part of teams of SWs and Nurse Practitioner extenders, including household health surveys in mental health and risk factors in needy communities, e.g., in low- income housing, as recently applied in NY State and California during COVID-19.
3. Appendix A needs a footnote including acronyms in the table.
4. Tables 1 and 2 need Titles.
5. Students in further programs would benefit from doing household surveys target populations of general as well as mental health.
6. Recommend submit revision
The additional comments:
1 The main question of the project should be restated and justified more clearly.
2 The topic is a global issue of mental health being dealt with in care giver training which is a legitimate research question, but I suggested addressing the context of future expansion of mental health monitoring of high risk communities in the New York State, USA, but which exist virtually everywhere in high-, medium- and low income countries..
3 This project adds self evaluation of efficacy of specific training of social workers and nurse practitioner in child/adolescent mental health issues in high risk population in communities. This contributes to the knowledge base of the topic of mental health care of adolescents.
4 It is not a randomized control research trial structure, and the limitations of the study should be clearly stated.
5 AS I indicated, the discussions and conclusions need strengthening with more discussion related to service models beyond classical models, such as strengthening outreach services by community health workers with similar training as “mental health care divers extenders” with home surveys of mental health cases for expansion of clinical care and telehealth methods applied in adolescent mental health case work. In other words, the training focus of this study must recognize the limitations of current models of services in this field, giving the example of community health workers (CHW) providing monitoring in other field of public health as seen in the Covid-19 CHW home visiting of high risk population groups in NY state.
6 I would suggest several references to back up the comments in point 5 above.
Author Response
In response to the comments, we
1) added the suggested citation. Thank you for the helpful current reference.
3) Added footnotes to Appendix A to include acronym references
4) added titles to Tables 1 and 2
2 and 5)
additional comments
1) This study was a program evaluation of a training program. This was clearly stated in the abstract and the introduction.
4) In lines 128-130, we added the following statement, "This was not a randomized control study and therefore, the findings are limited because there may have been selection bias in the students who chose to apply and participate in the program. In addition, we did not survey students who did not participate in the program on the study measures."
5) We discussed other methods of research in a new paragraph added lines 289-319.
Section 2 point 1) We restated the purpose of the research in the abstract lines 12-16.
point 2) We addressed other ways to conduct research in lines 289-319 which also responds to point 5 that we expand the discussion.